# Pharmacological prevention of bone loss and fractures following solid organ transplantations: Protocol for a systematic review and network meta-analysis

Jiawen Deng[1,2,3]* , Myron Moskalyk[4]* , Wenteng Hou[1,5], Qi Kang Zuo[6], Jinyu Luo[4]

1 Faculty of Health Sciences, McMaster University, Hamilton, ON, Canada, 2 Temerty Faculty of Medicine, University of Toronto, Toronto, ON, Canada, 3 Li Ka Shing Knowledge Institute, St. Michael's Hospital, Toronto, ON, Canada, 4 Biostatistics Division, Dalla Lana School of Public Health, University of Toronto, Toronto, ON, Canada, 5 Schulich School of Medicine & Dentistry, Western University, London, ON, Canada, 6 UBC Faculty of Medicine, University of British Columbia, Vancouver, BC, Canada

☯ These authors contributed equally to this work.
* dengj35@mcmaster.ca (JD); myron.moskalyk@mail.utoronto.ca (MM)

**Data Availability Statement:** No datasets were generated or analysed during the current study. All

# Abstract

## Introduction

Solid organ transplant (SOT) recipients can experience bone loss caused by underlying conditions and the use of immunosuppressants. As a result, SOT recipients are at risk for decreased bone mineral density (BMD) and increased fracture incidences. We propose a network meta-analysis (NMA) that incorporates all available randomized control trial (RCT) data to provide the most comprehensive ranking of anti-osteoporotic interventions according to their ability to decrease fracture incidences and increase BMD in SOT recipients.

## Methods

We will search MEDLINE, EMBASE, Web of Science, CINAHL, CENTRAL and CNKI for relevant RCTs that enrolled adult SOT recipients, assessed anti-osteoporotic therapies, and reported relevant outcomes. Title and full-text screening as well as data extraction will be performed in-duplicate. We will report changes in BMD as weighted or standardized mean differences, and fracture incidences as risk ratios. SUCRA scores will be used to provide rankings of interventions, and quality of evidence will be examined using RoB2 and CINeMA.

## Discussions

To our knowledge, this systematic review and NMA will be the most comprehensive quantitative analysis regarding the management of bone loss and fractures in SOT recipients. Our analysis should be able to provide physicians and patients with an up-to-date recommendation for pharmacotherapies in reducing incidences of bone loss and fractures associated with SOT. The findings of the NMA will be disseminated in a peer-reviewed journal.

relevant data from this study will be made available upon study completion.

**Funding:** The author(s) received no specific funding for this work.

**Competing interests:** The authors have declared that no competing interests exist.

## Introduction

Solid organ transplantation (SOT) has become the standard of care for patients living with end-stage organ failure and organ insufficiencies in recent years. According to the United Network for Organ Sharing (UNOS), there are over 110,000 patients awaiting life-saving organ transplants in the US, with over 46,000 SOT surgeries performed in 2023 [1]. These numbers are expected to increase significantly in the near future, with worrying public health trends—such as the continuously increasing rate of cardiac failures worldwide—showing no signs of slowing down [2].

With the increasing number of transplant recipients and improvements in survival rates as a result of advancements in surgical techniques, perioperative care, and immunosuppressive therapies, improving patients' postoperative recovery and long-term survival has become ever more important. SOT patients face many morbidity risks as a result of transplant-related consequences; the risk of osteoporosis for SOT patients, for example, is five times greater than that of the general population [3].

While a majority of bone loss associated with SOT occurs in the 6–12 months postoperative period [4], the risks of osteoporosis can increase in some SOT recipients even before the transplant. For example, abnormal bone formation and resorption markers are widely observed in liver transplant candidates, while altered bone metabolism is associated with hormonal abnormalities in chronic kidney disease patients [4]. In heart and lung transplant candidates, the pathophysiology and pharmacotherapies for diseases such as congestive heart failure and cystic fibrosis are also associated with an increased risk for osteoporosis pre-transplant [3,4].

After transplant, lifelong immunosuppression is a major causative factor in transplant-related osteoporosis. Glucocorticoids, a common immunosuppressant agent for SOT patients, activates the RANKL system, which decreases bone formation and accelerates bone resorption [4]. Other medications, such as cyclosporin A and other calcineurin inhibitors, greatly improve SOT survival by suppressing T cell activation and proliferation, but can also induce high-turnover bone loss through accelerated bone resorption [5].

Because SOT patients are at such high risk for bone loss after transplant, osteoporosis management is imperative. Calcium and vitamin D supplementation is recommended for all SOT patients starting before the transplant surgery, as 91% of all end-stage organ failure patients experience vitamin D insufficiencies [6]. Beside its graft-protective effects, vitamin D can also improve intestinal calcium absorption, prevent hyperparathyroidism, and promote osteoblast formation [3]. In addition, the effectiveness of calcitonin as an adjuvant to calcium and vitamin D has been demonstrated in renal and cardiac transplants [3]. Bisphosphonates have also been proven as robust anti-osteoporotic agents in SOT recipients. They maintain BMD in post-transplant patients by mediating osteoclast-related bone resorption [4] and some long-term studies have demonstrated that they decrease fracture risks in SOT patients [3].

Despite the vast amount of clinical trial evidence available to support the use of different anti-osteoporotic drugs in transplant patients, no systematic review to date has incorporated all available randomized controlled trial (RCT) data to determine the optimal anti-osteoporotic intervention for SOT recipients. Previous meta-analyses compared the efficacy of bisphosphonates and vitamin D analogs in increasing BMD and decreasing fracture incidences in SOT patients, however they were limited by the pairwise meta-analysis design and thus was not able to investigate the efficacy of different bisphosphonates or vitamin D analogs. In addition, they were not able to incorporate RCT data on other anti-osteoporotic interventions, such as calcitonin and novel antiosteoporotic therapies such as denosumab and teriparatide/abaloparatide [7].

Network meta-analysis (NMA) allows for the comparison and ranking of multiple interventions at once as opposed to regular, pairwise meta-analyses [8]. We propose to conduct a

systematic review and NMA of RCTs to investigate the following research questions: what are the comparative effects (i.e., fracture incidences and changes in BMD from baseline) of different anti-osteoporotic interventions on adult SOT recipients?

## Materials and methods

This protocol was developed according to the Preferred Reporting Items for Systematic Reviews and Meta-Analyses (PRISMA) for systematic review protocols (PRISMA-P) extension statement [9]. A PRISMA-P checklist is appended as **S1 Table**. We will conduct this systematic review and NMA in accordance with the Preferred Reporting Items for Systematic Reviews and Meta-Analyses incorporating NMA of health care interventions (PRISMA-NMA) [10] and the Cochrane Handbook for Systematic Reviews of Interventions [11]. This study is prospectively registered on PROSPERO (CRD42019138807) [12]. Significant amendments to this protocol will be reported and published alongside the results of the review.

### Outcome measures

Our primary outcome measure is the incidence of vertebral and nonvertebral fractures. Our secondary outcome measures will include: 1) changes in femoral neck, lumbar spine, and total hip BMD from baseline, 2) incidence of adverse events, and 3) incidence of serious adverse events (SAEs). Fracture was chosen as a patient-important efficacy outcome for bone loss and osteoporosis, while BMD is chosen as a widely-recognized and commonly reported surrogate measure of fracture risk, osteoporosis severity, and bone health [13–15]. We will evaluate all outcomes based on data collected at the latest follow-up visit. Fractures and adverse events will be defined as per individual study criteria.

### Eligibility criteria

We will include parallel-group or crossover RCTs that: 1) enrolled adult (age ≥18 years old) SOT recipients, 2) compared an anti-osteoporotic medication against placebo/no anti-osteoporotic treatment or against a different anti-osteoporotic treatment, and 3) reported any of our target outcomes. Studies assessing combinations of anti-osteoporotic medications are eligible. If a RCT used a crossover design, latest data from before the first crossover will be used. SOT is defined as transplantation of the kidney, liver, pancreas, heart, or lung [16].

We will include any anti-osteoporotic pharmacotherapies used for the purposes of preventing bone loss and fractures. This includes (but is not limited to): bisphosphonates (e.g., alendronate, risedronate, ibandronate, and zoledronic acid), calcitonin, calcium supplementation, vitamin D and D analogs, raloxifene, parathyroid hormones (i.e., teriparatide and abaloparatide), denosumab, and romosozumab.

### Study identification

We will conduct a database search in Ovid Medline, Ovid Embase, Web of Science (Core Collection), EBSCOhost CINAHL, Cochrane CENTRAL, and the China National Knowledge Infrastructure (CNKI) from inception to February 2nd, 2024. The search strategy was developed and piloted on Ovid MEDLINE using relevant MeSH terms and title/abstract headings such as "transplant recipient," "osteoporosis," "bone resorption," and "fractures" and a modified version of the Cochrane Highly Sensitive Search Strategy for identifying randomized trials in MEDLINE [11]. The search strategy was then translated to other databases using database-specific keywords. A sample search strategy is tabulated in **S2 Table**.

In addition to the database searches, we will also hand-search the reference list of previous systematic reviews, as well as clinical trial registries (i.e., ClinicalTrials.gov and the WHO International Clinical Trials Registry Platform) and preprint servers (i.e., medRxiv and Research Square) for relevant published data.

To account for potential changes in clinical practice and ensure that our findings are current, we will proactively monitor literature sources for new publications throughout the review process, and incorporate newly published data or conduct updated database searches, if needed.

## Study selection

We will perform title and abstract screening independently and in-duplicate using Covidence (https://www.covidence.org/) [17]. Records deemed eligible will be subsequently retrieved and entered into an in-duplicate full-text screening process. Disagreements will be resolved by discussions mediated by a senior author to reach consensus.

## Data extraction

We will perform data extraction independently and in-duplicate using standardized data extraction sheets developed *a priori*. Data from intention-to-treat (ITT) or modified ITT analyses will be preferentially extracted over data from per-protocol analyses. We will resolve discrepancies by recruiting a senior author to review the data.

## Data items

Data items that will be extracted include:

1. **Bibliometric data:** author names, year of publication, trial registration number, digital object identifier, publication journal, funding sources, and conflict of interest.

2. **Methodology descriptions:** number of participating centers, study setting, blinding methods, enrollment period, randomization and allocation methods, BMD measurement methods, and fracture definition.

3. **Treatment descriptions:** description of anti-osteoporotic therapies (i.e., name, type, dosage, and duration), description of adjuvant therapies, and description of placebos (if applicable).

4. **Baseline information:** number of patients randomized, number of patients analyzed for efficacy and safety outcomes, number of patients lost to follow-up, mean age and age variance, and sex distributions.

5. **Outcome data:** Mean/median baseline BMD measurements with variance, mean/median final BMD measurements with variance, mean/median absolute change in BMD from baseline with variance, mean/median percentage change in BMD from baseline with variance, incidence of vertebral fracture, incidence of non-vertebral fracture, incidence of adverse events, incidence of SAEs, follow-up duration of the latest efficacy and safety outcome measurements and treatment adherence at the latest follow-up.

All extracted data will be available to readers of the final review publication upon request.

## Risk of bias

We will assess risk of bias (RoB) independently and in-duplicate using the Revised Cochrane risk of Bias Tool for Assessing Randomised Trials (RoB2) [18]. The assessments will be aimed

at examining the effect of assignment to interventions, given our focus on extracting intention-to-treat outcome data. Disagreements will be resolved by discussions mediated by a senior author to reach consensus.

## Network meta-analysis

We will perform Bayesian NMAs using the *gemtc* and *BUGSnet* library [8] in *R* and Markov chain Monte Carlo engine JAGS [19]. Because we expect heterogeneity among studies due to differences in patient characteristics and study methodologies, we will use a random-effects model [8]. The NMAs will be stratified by SOT type. We will assume a vague prior for the between-study heterogeneity with uniform distribution. Patients receiving no active anti-osteoporotic interventions, including patients using placebo, will be used as a reference for comparison. Transitivity will be assessed qualitatively by comparing patients and study characteristics.

For changes in BMD, we will report the results of the analysis as mean differences with 95% credible intervals (CrIs) if all included studies utilized the same scale (e.g., if BMD changes are all reported as percentage changes). Otherwise, we will report these outcomes as standardized mean differences (SMDs) to include all available RCT data. We will calculate SMD by dividing the mean differences between treatment groups by the weighted pooled standard deviation (SD) using the DerSimonian-Laird method [20]. Fracture incidences will be reported as risk ratios with corresponding 95% CrIs. Model convergence will be assessed using the Brooks-Gelman-Rubin method as well as examination of trace and autocorrelation plots [21]. We will run all network models for a minimum of 100,000 iterations to ensure convergence. Network graphs will be provided to visualize the overall network structure of comparisons.

If there are outcomes for which we did not gather enough information to perform a NMA, we will provide a qualitative description of the available data and study outcomes or perform a pairwise meta-analysis if data permits.

## Heterogeneity and inconsistency assessment

We will assess between-study heterogeneity within each pairwise comparison using $I^2$ statistics [22]. Inconsistency will be assessed using the node-splitting method [23]. If inconsistency is identified, we will conduct network meta-regression models and subgroup analysis to identify how potential effect modifiers may impact our results. Based on previous literature reviews, our assessments for sources of inconsistency will be focused on age, sex, and BMI [24]. In addition, we will conduct one-stage sensitivity analysis with E-value analogs (if applicable) to evaluate the strength of unmeasured confounding [25] if we identified RCTs with a high risk of bias.

## Treatment ranking

We will use the surface under the cumulative ranking curve (SUCRA) scores to provide an estimate as to the ranking of treatments. SUCRA scores range from 0 to 1, with higher SUCRA scores indicating more efficacious treatment arms [26].

## Missing data

We will attempt to contact the authors of the original studies to obtain missing or unpublished data. For studies that did not report change in BMD but reported BMD values at baseline and follow up visits, we will impute the mean change and associated variances using correlation coefficients, as recommended by the Cochrane Handbook. For studies that reported median

and range or median and interquartile range as summary statistics for continuous measure-ments, we will first assess the normality of the data using methods proposed by Shi et al. [27]. If the data is not significantly skewed, we will impute mean and standard deviation using methods proposed by Shi et al. [28], Luo et al. [29], and Wan et al. [30].

## Publication bias

We will use comparison-adjusted funnel plots to assess for small-study effects within networks as indications for publication bias [31]. As the treatment arms need to be sorted in a meaning-ful way to generate comparison-adjusted funnel plots, we will order the treatment arms to define each comparison as an intervention versus a control treatment, with the assumption that publication bias is likely to exaggerate the effectiveness of the intervention treatment [32]. Egger's regression test will be used to examine the presence of asymmetry in the comparison-adjusted funnel plots [33].

## Quality of evidence

We will use the Confidence in Network Meta-Analysis (CINeMA) web application (https://cinema.ispm.unibe.ch/) [34] to evaluate confidence in the findings from our NMA. The CINeMA framework is similar to the Grading of Recommendations, Assessment, Development, and Evalu-ations (GRADE) framework [35,36] and evaluates network quality based on six criteria: within-study bias, across-study bias, indirectness, imprecision, heterogeneity and incoherence [37]. We will summarize the results and quality of our NMA using a summary of findings table.

## Ethics and dissemination

No ethics approval is needed as only published data will be used and analyzed. The completed review will be submitted for publication in a peer-reviewed journal.

## Discussions

Previous meta-analyses regarding the use of anti-osteoporotic medications in SOT recipients were limited by their pairwise designs. Our study aims to significantly expand upon previous reviews by incorporating the entirety of RCT evidence available.

   Our proposed review will have several strengths. First, we will extend our database search to Chinese databases for our analysis. Because of China's immense patient population and reg-ulations that promote pharmaceutical research, the inclusion of Chinese RCTs will help strengthen the power and precision of our analyses [38]. Furthermore, we will use NMA tech-niques to analyze RCTs assessing anti-osteoporotic pharmacotherapies. This strategy will allow us to include different bisphosphonates as separate treatment arms, while including other interventions such as calcitonin. Lastly, we will only include RCT data, and we will use tools such as RoB2 and CINeMA to evaluate the quality of our included studies and networks.

   Our NMA will likely be the largest quantitative synthesis assessing anti-osteoporotic thera-pies among SOT recipients to date. The proposed review should help physicians with selecting anti-osteoporotic regimens that are the most beneficial for the bone health of SOT recipients. Our study may also highlight promising treatment strategies that were not discussed in previ-ous analyses, providing physicians and researchers with future research directions.

## Limitations

There are several potential limitations to consider in this proposed study. Firstly, we expect to include a clinically heterogeneous patient population, even after stratifying by SOT types. We

aim to assess and mitigate the impact of clinical confounders through the use of node-splitting, meta regressions, and sensitivity analyses. Secondly, there may be certain differences in clinical practices adopted by RCTs originating from different countries that may impact our findings and contribute to higher heterogeneity. However, we expect to account for this by seeking expertise from our multinational and multilingual research team. This effect may also be mitigated by the existence of standardized international clinical practice guidelines for post-SOT care, which reduces the diversity in clinical practices among different geographical regions [39]. Finally, our analysis may be limited by sparse literatures surrounding the use of prior distributions in this field. We will be using vague priors to allow the data to inform the posterior distribution more heavily.

## Supporting information

**S1 Table. PRISMA-P checklist.**
(DOCX)

**S2 Table. Ovid MEDLINE search strategy.**
(DOCX)

## Author Contributions

**Conceptualization:** Jiawen Deng, Wenteng Hou, Qi Kang Zuo.

**Methodology:** Jiawen Deng, Myron Moskalyk, Jinyu Luo.

**Project administration:** Jiawen Deng.

**Resources:** Jiawen Deng.

**Supervision:** Myron Moskalyk.

**Writing – original draft:** Jiawen Deng, Myron Moskalyk, Wenteng Hou.

**Writing – review & editing:** Jiawen Deng, Myron Moskalyk, Wenteng Hou, Qi Kang Zuo, Jinyu Luo.

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
