## [Decision Letter · Decision Letter 0]

28 Feb 2024

PONE-D-24-04602Pharmacological prevention of bone loss and fractures following solid organ transplantations: Protocol for a systematic review and network meta-analysisPLOS ONE

Dear Dr. Deng,

Thank you for submitting your manuscript to PLOS ONE. After careful consideration, we feel that it has merit but does not fully meet PLOS ONE’s publication criteria as it currently stands. Therefore, we invite you to submit a revised version of the manuscript that addresses the points raised during the review process.

We look forward to receiving your revised manuscript.

Kind regards,

Melissa Orlandin Premaor, M.D., Ph.D

Academic Editor

PLOS ONE

Journal Requirements:

Reviewers' comments:

Reviewer's Responses to Questions

**Comments to the Author**

1. Does the manuscript provide a valid rationale for the proposed study, with clearly identified and justified research questions?

Reviewer #1: Yes

Reviewer #2: Yes

2. Is the protocol technically sound and planned in a manner that will lead to a meaningful outcome and allow testing the stated hypotheses?

Reviewer #1: Yes

Reviewer #2: Yes

3. Is the methodology feasible and described in sufficient detail to allow the work to be replicable?

Reviewer #1: Yes

Reviewer #2: Yes

4. Have the authors described where all data underlying the findings will be made available when the study is complete?

Reviewer #1: No

Reviewer #2: No

5. Is the manuscript presented in an intelligible fashion and written in standard English?

Reviewer #1: Yes

Reviewer #2: Yes

6. Review Comments to the Author

You may also provide optional suggestions and comments to authors that they might find helpful in planning their study.

Reviewer #1: How do you plan to address potential confounding factors in your study, such as differences in patient demographics, comorbidities, and medication histories across the included trials?

Could you elaborate on the rationale behind your choice of outcome measures, particularly in terms of their clinical relevance and sensitivity to detecting differences between treatment groups?

How will your study account for potential changes in clinical practice and the availability of new evidence during the analysis phase?

In your discussion, you mentioned the importance of incorporating Chinese RCTs into your analysis. Could you discuss any challenges you anticipate in accessing and interpreting data from these studies, particularly in terms of language barriers and cultural differences in clinical practice?

Could you discuss the potential limitations of your study?

The study protocol does not explicitly mention where all data underlying the findings will be made available when the study is complete. To ensure transparency and facilitate reproducibility, it would be advisable for the authors to explicitly state their plans for data sharing and accessibility in the final publication or in subsequent updates to the study protocol.

Reviewer #2: Minor suggestions:

- It is preferable to mention which are your primary outcome(s) and which ones are your secondary outcomes.

- There is no mention of how articles published in languages other than English will be handled. This is particularly relevant for your study since you are searching a Chinese database.

- I suggest the addition of network graphs to visualize the overall structure of the comparisons in your network.

- I am not familiar with this field of research, but I am wondering if treatment retention would be important information to extract and report.

7. PLOS authors have the option to publish the peer review history of their article (what does this mean?). If published, this will include your full peer review and any attached files.

Reviewer #1: **Yes: **Daniela A. Rodrigues

Reviewer #2: No

---

## [Author Response · Author response to Decision Letter 0]

12 Mar 2024

Please see the uploaded "Response to Reviewer Comment" document.

---

## [Editor Report · Decision Letter 1]

8 Apr 2024

Pharmacological prevention of bone loss and fractures following solid organ transplantations: Protocol for a systematic review and network meta-analysis

PONE-D-24-04602R1

Dear Dr. Deng,

We’re pleased to inform you that your manuscript has been judged scientifically suitable for publication and will be formally accepted for publication once it meets all outstanding technical requirements.

Kind regards,

Melissa Orlandin Premaor, M.D., Ph.D

Academic Editor

PLOS ONE